# Potential of Enzymatically Synthesized Hemozoin Analog as Th1 Cell Adjuvant

**DOI:** 10.3390/nano14171440

**Published:** 2024-09-03

**Authors:** Kazuaki Hoshi, Anh Thi Tram Tu, Miwako Shobo, Karin Kettisen, Lei Ye, Leif Bülow, Yoji Hakamata, Tetsuya Furuya, Ryutaro Asano, Wakako Tsugawa, Kazunori Ikebukuro, Koji Sode, Tomohiko Yamazaki

**Affiliations:** 1Research Center for Macromolecules and Biomaterials, National Institute for Materials Science (NIMS), Tsukuba 305-0047, Japantttanh@hcmus.edu.vn (A.T.T.T.); shobo.miwako@nims.go.jp (M.S.); 2Department of Magnetic and Biomedical Materials, Faculty of Materials Science and Technology, University of Science, Ho Chi Minh City 70000, Vietnam; 3Ho Chi Minh City Campus, Vietnam National University, Ho Chi Minh City 70000, Vietnam; 4Division of Pure and Applied Biochemistry, Department of Chemistry, Lund University, 22100 Lund, Sweden; karinkettisen@gmail.com (K.K.); lei.ye@tbiokem.lth.se (L.Y.); 5School of Veterinary Nursing and Technology, Nippon Veterinary and Life Science University, Musashino 180-8602, Japan; yhakama@nvlu.ac.jp; 6Cooperative Department of Veterinary Medicine, Faculty of Agriculture, Tokyo University of Agriculture and Technology, Fuchu 183-8509, Japan; furuyat@cc.tuat.ac.jp; 7Department of Biotechnology and Life science, Graduate School of Engineering, Tokyo University of Agriculture and Technology, Koganei 184-8588, Japan; ryutaroa@cc.tuat.ac.jp (R.A.); tsugawa@cc.tuat.ac.jp (W.T.); ikebu@cc.tuat.ac.jp (K.I.); 8Joint Department of Biomedical Engineering, The University of North Carolina at Chapel Hill and North Carolina State University, Chapel Hill, NC 27599, USA; ksode@email.unc.edu; 9Graduate School of Life Science, Hokkaido University, Sapporo 060-0808, Japan

**Keywords:** hemozoin analog, heme detoxification protein, adjuvant, vaccine, Th-1 immunity

## Abstract

Hemozoin (Hz) is a heme crystal produced during malaria infection that stimulates immune cells, leading to the production of cytokines and chemokines. The immunostimulatory action of Hz has previously been applied in the development of alternative adjuvants. Crystallization of hemin is a chemical approach for producing Hz. Here, we focused on an enzymatic production method for Hz using the heme detoxification protein (HDP), which catalyzes heme dimer formation from hemin in *Plasmodium*. We examined the immunostimulatory effects of an enzymatically synthesized analog of Hz (esHz) produced by recombinant *Plasmodium falciparum* HDP. Enzymatically synthesized Hz stimulates a macrophage cell line and human peripheral mononuclear cells, leading to the production of interleukin (IL)-6 and IL-12p40. In mice, subcutaneous administration of esHz together with an antigen, ovalbumin (OVA), increased the OVA-specific immunoglobulin (Ig) G2c isotype level in the serum, whereas OVA-specific IgG1 was not induced. Our findings suggest that esHz is a useful Th-1 cell adjuvant.

## 1. Introduction

Heme detoxification protein (HDP) is a heme ligase that plays an essential role in the formation of the malarial pigment hemozoin (Hz) during the erythrocytic stage of *Plasmodium* parasites. *Plasmodium* digests hemoglobin, which is abundant in erythrocytes, to obtain nutrients essential for its growth. This process is accompanied by the release of free heme, which is toxic to *Plasmodium* because it oxidizes parasitic DNA and lipids [1]. During the degradation of hemoglobin, HDP binds to toxic-free heme and converts them to non-toxic heme crystals, such as Hz. Heme detoxification protein was first identified in *Plasmodium falciparum*, and orthologs have been found in seven other *Plasmodium* species [2]. The HDP reaction mechanism was studied using recombinant HDP [2,3,4,5]. A previous report indicated that HDP is the most potent enzyme for converting heme to Hz among other available Hz producers, such as lipids and histidine-rich protein II.

Natural Hz is not only a metabolite but also activates immune cells during malaria infection. After heme detoxification, Hz is released into the bloodstream upon the rupture of erythrocytes and subsequently accumulates in the reticuloendothelial system of the host (i.e., macrophages, leukocytes, and tissues, such as the liver and spleen), resulting in an inflammatory response by the host’s immune system [6,7,8,9,10,11,12,13,14,15]. It induces the production of cytokines and chemokines, such as interleukin (IL)-6, IL-12, IL-8, tumor necrosis factor-α, and monocyte chemoattractant protein 1 from dendritic cells and macrophages [7]. Accumulation of Hz in tissues is associated with disease severity in malaria-infected patients and mice [7,16,17].

Innate immune receptors, such as Toll-like receptors (TLRs) [6], C-type lectin receptors [18], and nucleotide-binding oligomerization domain-like receptors (NLRs) [8], may recognize natural Hz and mediate immune responses. A chemically synthesized analog of Hz (csHz) is produced from hemin using an acid-catalyzed method and is structurally identical to natural Hz comprising heme dimers [19]. The generated csHz can also induce the production of pro-inflammatory cytokines, such as IL-6, IL-1α, and IL-1β [8]. 

Natural Hz and csHz were prepared by recovery from erythrocytes infected with *Plasmodium* and crystallization of chemically produced hemin, respectively. However, there are no reports on the application of Hz synthesis in vitro using an enzymatic HDP reaction. The conversion ratio of heme to Hz by recombinant *Plasmodium falciparum* HDP (PfHDP) was comparable to that of native HDP [2]. Therefore, the enzymatic reaction of PfHDP may be applicable for the preparation of Hz analogs that function as adjuvant molecules.

Here, we hypothesized that an enzymatically synthesized analog of Hz (esHz) produced by recombinant PfHDP using hemin as a substrate could have immunostimulatory effects and act as an adjuvant, similar to csHz [7]. 

The esHz synthesized by recombinant HDP showed a hydrodynamic size of 350 nm, which is similar to that of natural Hz. Enzymatically synthesized Hz induced 27- and 5-fold higher levels of IL-6 and interferon (IFN)-γ, respectively, in human peripheral blood mononuclear cells (PBMCs) than csHz. Subcutaneous administration of esHz together with the model antigen ovalbumin (OVA) increased antigen-specific immunoglobulin (Ig) G2c production in the blood six-fold compared to alum with OVA at six weeks after the second vaccination. Additionally, esHz tended to induce OVA-specific IgG2c more selectively than IgG1, resulting in the induction of a Th-1 cell immune reaction. This study clearly indicates that esHz produced by recombinant PfHDP can be useful for the production of a vaccine adjuvant to induce Th-1 cell immunity.

## 2. Materials and Methods

### 2.1. Reagents

Chaperone-competent cells, pG-Tf2/BL21 and pCold IV, for cold shock-induced protein expression in *E. coli* were purchased from Takara Bio (Shiga, Japan). Hemin was obtained from the Tokyo Chemical Industry (Tokyo, Japan). HisTrap^TM^ HP and HiTrap^TM^ Q HP columns were acquired from Cytiva (Marlboro, MA, USA). His-tagged monoclonal antibodies were purchased from Novagen (Madison, WI, USA). Horseradish peroxidase (HRP)-conjugated goat anti-mouse IgG was obtained from DakoCytomation (Glostrup, Denmark). Phosphorothioate-modified CpG ODN 2006 (5′-TCGTCGTTTTGTCGTTTTGTCGTT-3′) was synthesized by Fasmac (Kanagawa, Japan). Phosphorothioate-modified CpG ODN K3 (5′-ATCGACTCTCGAGCGTTCTC-3′) was purchased from GeneDesign (Osaka, Japan).

### 2.2. Expression and Purification of HDP

To express HDP with a hexahistidine tag at the C-terminus, synthetic, full-length, codon-optimized DNA of PfHDP (GenBank Acc# NP_702335) was synthesized by Eurofins Genomics (Tokyo, Japan). The HDP gene fragment was inserted into the HindIII/NdeI site of the pCold IV vector to generate pCold IV-PfHDP. *Escherichia coli* pG-Tf2/BL21 competent cells express GroEL, GroES, and trigger factor (Tf) under the control of the tetracycline-inducible promoter (Pzt-1). Competent cells transformed with either pCold IV or pCold IV-PfHDP were grown in 100 mL of Luria–Bertani liquid medium supplemented with 20 µg/mL chloramphenicol and 100 µg/mL carbenicillin, with or without the addition of 10 ng/mL tetracycline. *Escherichia coli* cells were cultured at 37 °C until the OD590 reached 0.4, and they were then incubated at 15 °C for 30 min. Isopropyl-β-D-thiogalactopyranoside (IPTG, 0.1 mM) was subsequently added to the medium, and the cells were cultured at 15 °C for another 24 h. The PfHDP was purified as described previously, with slight modifications [2]. The cell pellet was sonicated in 50 mM N-cyclohexyl-3-aminopropanesulfonic acid (CAPS) buffer (pH 11.0) containing 0.3% (*w*/*v*) N-lauroylsarcosine and 0.3 M NaCl. The soluble proteins were separated by centrifugation at 12,000× *g* at 4 °C for 20 min. The soluble proteins were applied to a HisTrap^TM^ HP column equilibrated with 50 mM CAPS (pH 11.0) containing 0.3% (*w*/*v*) N-lauroylsarcosine and 0.3 M NaCl and eluted with a linear imidazole gradient ranging between 0 and 200 mM in the same buffer. The fractions containing PfHDP were dialyzed against 50 mM CAPS (pH 11.0) containing 0.3% (*w*/*v*) N-lauroylsarcosine and applied to a HiTrap^TM^ Q HP column. The proteins were eluted with a linear NaCl gradient in 50 mM CAPS (pH 11.0) containing 0.3% (*w*/*v*) N-lauroylsarcosine. Fractions containing HDP were dialyzed against 25 mM CAPS (pH 11.0) containing 135 mM NaCl. The purity of the eluted proteins was examined by sodium dodecyl sulfate-polyacrylamide gel electrophoresis (SDS-PAGE). Histidine-tagged PfHDP was detected by Western blotting with a monoclonal antibody against the His-tag and polyclonal goat anti-mouse immunoglobulins/HRP. The concentration of the purified PfHDP was determined using a BCA protein assay kit (Thermo Fisher Scientific, Waltham, MA, USA).

### 2.3. esHz Formation Assay

The Hz formation assay used to quantify esHz was performed as previously described [4]. The PfHDP concentration for the assay was estimated using the BCA protein assay kit, and an estimated extinction coefficient of 36.0 mM^−1^ cm^−1^ for sample absorbance at 280 nm [4]. The PfHDP (0.13 mg, 5 µM) was incubated with hemin (0.39 mg, 600 µM) in 0.5 M sodium acetate (pH 4.8) at 37 °C for 1 h on an orbital shaker at 500 rpm. Lysozyme (0.13 mg) was used as a negative control for PfHDP. The reaction was stopped by adding 0.1% (*w*/*v*) SDS. The pellet was collected by centrifugation at 15,000× *g* for 10 min and washed three times with 2.5% (*w*/*v*) SDS and 0.1 M sodium bicarbonate (pH 9.1). After washing three times with Milli-Q water, the pellet was lyophilized. To determine the concentration of hemin converted by PfHDP, the washed pellet was dissolved in 0.1 M NaOH. The concentration of hemin was calculated by using an extinction coefficient of 58.4 mM^−1^ cm^−1^ by monitoring the absorbance at 385 nm [4].

### 2.4. csHz Preparation

A previously described protocol was used to synthesize csHz [11]. Briefly, hemin (45 mg) was dissolved in 4.5 mL of 1 N NaOH and mixed with 450 µL of 1 N HCl in a flask. The flask was placed in an oil bath heated at 60 °C, and 10.2 mL of 1 M sodium acetate (pH 4.8) was gradually added with stirring. After centrifugation at 12,000× *g* for 20 min, the pellet was washed three times with 2.5% (*w*/*v*) SDS, followed by the washing with 0.1 M sodium bicarbonate (pH 9.1) four times. The pellet was lyophilized for further analysis.

### 2.5. Generation of Infra-Red (IR) Spectra and Scanning Electron Microscope (SEM) Images

Attenuated total reflection Fourier transform IR (ATR-FTIR) spectroscopy was used to characterize the esHz. The ATR-FTIR spectra of esHz, csHz, and hemin were measured using an IRTracer-100 (Shimadzu, Kyoto, Japan) equipped with an ATR accessory (Quest ATR; Specac, Kent, UK). Images of the pellets were obtained by SEM (S-4800, Hitachi High-Tech, Tokyo, Japan). The acceleration voltage was set at 12 kV. The size distribution was analyzed by dynamic light scattering (DLS-8000, Otsuka Electronics, Osaka, Japan) using a He-Ne laser (633 nm).

### 2.6. Mammalian Cell Culture and Immunostimulation

The murine macrophage cell line (RAW264) was purchased from the RIKEN BioResource Center (Tsukuba, Japan). The RAW264 cells were cultured in minimum essential medium (MEM) supplemented with 10% (*v*/*v*) fetal bovine serum, 100 unit/mL penicillin, 100 μg/mL streptomycin (P/S), and 1% (*v*/*v*) non-essential amino acid solution (Wako Pure Chemical Industries, Osaka, Japan). The cells were seeded in 96 well-plates at 5 × 10^4^ cells/well and cultured at 37 °C in a humidified incubator with 5% CO_2_ for 24 h. The cells were incubated with csHz, esHz (50 µg/mL), phosphorothioate-modified CpG ODNs (4 µM), and sterilized deionized water as a control for 24 h. Commercially available frozen PBMCs were purchased from Cellular Technology Limited (Shaker Heights, OH, USA). The PBMCs were thawed according to the manufacturer’s instructions, seeded in a 96-well plate at a density of 1 × 10^6^ cells/well, and stimulated with csHz, esHz (50 µg/mL), phosphorothioate-modified CpG ODN K3 (1 µM), or sterilized deionized water at 37 °C in a humidified incubator containing 5% CO_2_ for 48 h. After the stimulation, the culture medium was collected. The levels of induced cytokine mRNAs and secreted cytokines were evaluated using real-time reverse transcription-quantitative polymerase chain reaction (RT-qPCR) and enzyme-linked immunosorbent assay (ELISA), respectively.

### 2.7. RT-qPCR and ELISA

The relative transcript levels of cytokines in the RAW264 cells were examined using RT-qPCR. Total RNA was extracted from the cells using ISOGEN (Nippon Gen, Tokyo, Japan) following the manufacturer’s instructions. Before being converted to cDNA by reverse transcriptase (Takara Bio), extracted RNA underwent a genomic DNA digestion step with RNase-Free DNase (Takara Bio). Interleukin-6, IL-12, and IFN-β transcripts were measured using RT-qPCR conducted on LightCycler^®^ 480 System II (Roche, Basel, Switzerland) using SYBR Green MasterMix (Roche) and the primers listed in Table 1. Amplification was conducted with 50 ng cDNA in a total reaction volume of 15 μL for 45 cycles, 10 s each at 95 °C, 60 °C, and 72 °C. After the PCR, the “crossing point” (Cp) value, that is, the cycle at which the fluorescence of a sample becomes significantly higher than the background fluorescence, was determined. The mRNA expression levels were normalized to those of glyceraldehyde 3-phosphate dehydrogenase, a housekeeping gene. The specificity of the amplified products was determined by analyzing their melting curves. The levels of IL-6 and IFN-γ secreted into the medium collected from PBMCs were determined using Ready-Set-Go! The ELISA kits (Thermo Fisher Scientific) were used according to the manufacturer’s instructions.

### 2.8. Administration of OVA with esHz in Mice and Measurement of OVA-Specific IgG Level

Male C57BL/6 mice (Charles River Laboratories Japan, Yokohama, Japan) were housed in a pathogen-free environment for one week before the start of the experiment. All protocols were approved by the Animal Care and Use Committee of the National Institute for Materials Science according to the Guidelines for Proper Conduct of Animal Experiments established by the Science Council of Japan. For in vivo immunostimulatory experiments, 9-week-old mice were injected intraperitoneally with 500 µg of either alum or esHz following the serum collection at 4 h after injection. For vaccination, 200 µL of a solution containing OVA (200 µg) and with either alum (500 µg) or esHz (500 µg) was injected subcutaneously into the back of 6-week-old mice. Vaccination was conducted twice at one-week intervals, and the mice were boosted with OVA (200 µg) at six weeks after the second dose. Blood samples were collected every other week from the vascular bundle just above the posterior mandible using an animal lancet (MEDIpoint, Mineola, NY, USA). Blood was placed in Torch tubes (Satokasei, Tochigi, Japan) containing coagulation reagent and separation gel, and serum was separated by centrifugation at 3000× *g* for 10 min at 4 °C. Serum was stored at −20 °C until ELISA. Ovalbumin-specific levels of IgG1, IgG2c, and total IgE were measured using ELISA as previously described [20].

### 2.9. Statistical Analysis

Statistical significance was evaluated using a one-way analysis of variance, followed by Tukey’s multiple comparison test. All statistical analyses were performed using GraphPad Prism version 8.2.0 Windows (GraphPad Software, La Jolla, CA, USA).

## 3. Results

### 3.1. Cold-Shock with Co-Expression of Chaperones Improves the Solubility of PfHDP

Figure 1A summarizes the expression and purification of PfHDP. In a previous study, recombinant HDPs were prepared from inclusion bodies via refolding [2]. Here, we attempted to express PfHDP using the cold shock expression method with the co-expression of chaperones to improve the solubility of PfHDP. Figure 1B shows the expression levels of PfHDP in soluble and insoluble fractions. The thick bands between the molecular weight markers of 47,300 and 84,700 Da correspond to the co-expressed Tf (~56,000) and GroEL (~60,000) proteins, respectively. His-tagged PfHDP has a molecular weight of approximately 25,000 Da [2]. The co-expression of chaperones increased the solubility of PfHDP in contrast to cold shock expression alone (Figure 1C). The expressed PfHDP was purified using Ni-NTA affinity chromatography followed by anion-exchange chromatography. After treating the samples with dithiothreitol, the purity of each eluted fraction was examined using SDS-PAGE (Figure 1D). A band corresponding to PfHDP was observed in the fraction eluted by anion-exchange chromatography. Finally, 0.83 mg of the purified PfHDP was obtained from 200 mL of culture medium.

The ability of PfHDP to convert heme into esHz was evaluated using a Hz formation assay. An aliquot of 5 µM PfHDP was mixed with 600 µM hemin at 37 °C in 0.5 M sodium acetate (pH 4.8) for 1 h. The esHz was purified by washing three times with 2.5% (*w*/*v*) SDS, 0.1 M sodium bicarbonate (pH 9.1), and Milli-Q water to remove PfHDP and unreacted hemin. A black precipitate was obtained (Figure 2A).

The pellet was then subjected to IR spectroscopy (Figure 2B). The IR spectra of esHz and commercial csHz showed peaks at 1206 and 1661 cm^−1^, corresponding to the C–O and C=O stretching vibrations of the heme carboxylate group, respectively [5]. These peaks are absent in the hemin spectra. The amount of hemin converted by PfHDP was 202 ± 20 µM (Figure 2C). Purified HDP converted 34% hemin into esHz. This is in agreement with a previous study, which showed that refolded PfHDP (5 µM) can convert almost 30% hemin into Hz [4]. The size distribution of esHz was analyzed by dynamic light scattering, which showed a single peak around 350 ± 160 nm (polydispersity index: 0.23), as demonstrated in Figure 2D. The SEM images at esHz are shown in Figure 2E. These results verified that soluble PfHDP has similar potency in converting hemin into Hz as refolded PfHDP, as shown previously [4,5].

### 3.2. esHz Induces Immunostimulatory Cytokine Production in RAW264 Cells and PBMCs

The immunostimulatory effects of esHz were evaluated in RAW264 cells and human PBMCs. Enzymatically synthesized Hz-activated RAW264 cells to induce the transcription of IL-6, IL-12, and IFN-β (Figure 3A). The mRNA levels of IL-6 and IL-12 induced by esHz were almost 16- and 5-fold higher than those induced by csHz, respectively. We also examined the effects of esHz on primary human immune cells (PBMCs). The levels of IL-6 and IFN-γ induced were significantly higher than those induced by csHz or by CpG-ODN K3, an agonist of TLR9, in PBMCs (Figure 3B). CpG-ODN K3 and CpG ODN 2006 are phosphorothioate-modified CpG-ODNs, which are potent adjuvants. These results indicate that esHz significantly induced the production of immunostimulatory cytokines in both murine RAW264 cells and human PBMCs.

### 3.3. esHz Induces the Cytokines and OVA-Specific IgG Induction in Mice

We first evaluated whether esHz alone stimulated innate immunity in mice. Alum, which is already used as an adjuvant in several vaccines, was used as a control. In previous reports, maximum cytokine production was observed 2–4 h after the injection of TLR9 agonists in mice [15,21]. Therefore, mouse blood was collected 4 h after esHz or alum administration, and cytokine levels were quantified. Enzymatically stimulated analog of Hz treatment induced the production of IL-6 (Figure 4A) and IL-12p40 (Figure 4B). Particularly, the level of IL-6 induced by esHz was higher than that induced by alum alone. Next, we examined the adjuvant effect of esHz on OVA as an antigen in mice and measured the levels of OVA-specific IgG1 and IgG2c in the serum. Figure 5 shows the OVA-specific IgG1 and IgG2c levels in the serum at two and six weeks after the second vaccination and two weeks after the boost. Vaccination with alum, an adjuvant molecule in practical use, induced OVA-specific IgG1 expression at all periods. In contrast, the OVA-specific IgG2c levels in the serum of mice injected with esHz as an adjuvant were higher than those in mice injected with OVA alone or a mixture of OVA and alum. Mice vaccinated with esHz as an adjuvant showed higher levels of OVA-specific IgG2c than those vaccinated with alum as an adjuvant at all time points after the second vaccination. Ovalbumin-specific IgG2c levels in mice vaccinated with a mixture of OVA and esHz gradually increased until six weeks after the second dose (Appendix A). A significant increase in OVA-specific IgG2c levels in the serum of mice vaccinated with esHz as an adjuvant was also observed in a booster experiment with OVA after vaccine administration.

The dose ratio of OVA-specific IgG2c to OVA-specific IgG1 in mice vaccinated with esHz as an adjuvant was 28- and 20-fold higher than that in mice vaccinated with alum at six and two weeks after the second vaccination and boost, respectively. The results showed that esHz induced OVA-specific IgG2c more selectively than OVA-specific IgG1. Additionally, esHz vaccination was not accompanied by the induction of OVA-specific IgE, an indicator of vaccination side effects (Appendix A). These results indicated that esHz induces Th-1 cell immune responses.

## 4. Discussion

This study examined the immunostimulatory effects of esHz produced using recombinant PfHDP. In the present study, we demonstrated that esHz stimulated innate immunity in mice and induced antigen-specific IgG2c. To our knowledge, this is the first study to demonstrate that esHz can function as a Th-1 adjuvant.

Previous studies have used either the pET or pCold expression systems to produce PfHDPs in the form of inclusion bodies [2,3,4]. Although both systems produced small amounts of solubilized PfHDP, the expression level was not high enough to allow for further purification [2,4]. The PfHDP was refolded in CAPS buffer (pH 11.0) containing L-arginine-HCl and oxidized glutathione [2]. In contrast, a previous study clearly indicated that co-expression of chaperones promoted the solubilization of PfHDP in *E. coli* cells cultured at 37 °C [22]. In this study, we examined the effects of cold shock on the co-expression of chaperones. This strategy improved the solubility of PfHDP compared to cold shock expression alone (Figure 1B). A total of 0.83 mg of the PfHDP was prepared from 200 mL of the *Escherichia coli* culture. This high yield of soluble recombinant PfHDP from *Escherichia coli* has not been previously reported.

Hemozoin was conducted in the digestive vacuole of Plasmodium under acidic conditions. The particle size of natural Hz varies among *Plasmodium* species [23]. All four *Plasmodium falciparum* clones produced natural Hz with average dimensions of approximately 100 × 100 × 300–500 nm [23]. Natural Hz is brick-like with smooth sides at right angles and shows much less size heterogeneity [23]. In contrast, csHz produced by hemin crystallization differs morphologically from natural Hz. The size of csHz has a wider distribution than natural Hz, with particle sizes in the range of 50 nm to 20 µm [6]. The csHz has tapered ends and radiates in different directions [23]. The esHz was brick-like with smooth sides (Figure 2D). The crystal had a size of approximately 350 nm (Figure 2C), which consisted of heme dimers (Figure 2A). The morphology of the esHz was more similar to natural Hz than that of the csHz. Additionally, esHz showed stronger immunostimulatory effects than csHz (Figure 3). Therefore, esHz is an analog molecule of Hz that possesses morphological and immunostimulatory characteristics similar to those of natural Hz. The differences in immunostimulatory properties between natural Hz and csHz remain controversial. The crystal size of csHz is dependent on synthetic methods and tends to cause a diverse distribution of its size from 50 to 20 µm [6]. In contrast, DLS analysis of esHz showed a narrower distribution of crystal size from 100 nm to 1 µm (Figure 2C). The crystal size of Hz might be the reason for the difference in immunostimulatory properties (Figure 3). Several innate immune receptors, such as TLRs and NLRs, are involved in the recognition of natural Hz [6,8,18]. Heme, a component of Hz, stimulates TLR4 signaling [24]. Based on these reports, esHz may stimulate these innate immune receptors and exert immunostimulatory effects.

The adjuvant effects of csHz have been demonstrated in several animal models, with crystals in the range of 200 nm exerting the highest effects [25]. In mice, csHz induces antigen-specific IgG production and may act as a type 2 adjuvant. In contrast, in dog and monkey models, csHz may initiate Th-1-like immune responses against malaria and allergens [6,25]. The esHz stimulated immunity and induced IL-6 and IL-12p expression (Figure 4), resulting in the production of OVA-specific IgG (Figure 5). Particularly, the level of OVA-specific IgG2 induced by esHz was higher than that induced by alum (Figure 5). The potential of esHz as a Th-1 adjuvant was demonstrated by the lack of OVA-specific IgE induction (Appendix A). These results indicated that esHz can function as a Th-1 adjuvant.

## 5. Conclusions

We demonstrated the immunostimulatory effects and potential of esHz as a Th-1 adjuvant. The esHz was enzymatically synthesized using the soluble form of recombinant PfHDP. In terms of its morphology and immunostimulatory effects, esHz has properties similar to those of natural Hz. The enzymatic system based on recombinant PfHDP is a safe synthetic method of Hz adjuvant. The immunostimulatory effect of esHz was significantly higher than that of csHz in both mouse Raw264 macrophage cells and human PBMCs. In vivo experiments confirmed that esHz induces antigen-specific antibodies in mice. In particular, the antigen-specific IgG2c levels induced by esHz were significantly higher than those induced by alum. We demonstrated that recombinant PfHDP is a useful enzyme for producing a natural vaccine adjuvant, esHz, which induces Th-1 cell immunity.

## Figures and Tables

**Figure 1 nanomaterials-14-01440-f001:**
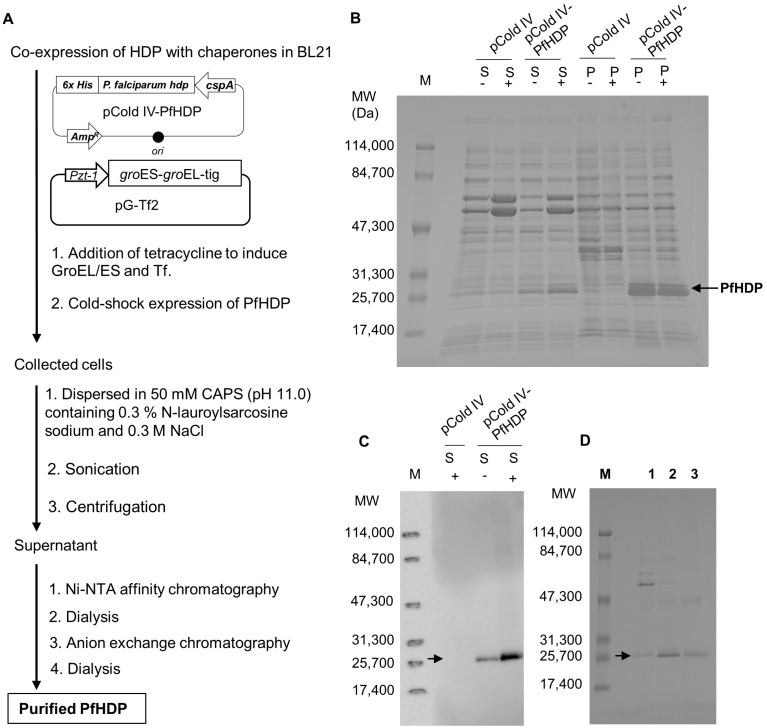
Expression and purification of PfHDP in *E. coli*. (**A**) Schematic diagram of PfHDP expression in *E. coli* and purification by chromatography. The BL21 cells co-transfected with pCold IV-PfHDP and pG-TF were grown in the medium either without (-) or with (+) the addition of tetracycline, followed by protein induction by cold shock expression. The expression levels of PfHDP in either the soluble (S) or insoluble (P) fraction were analyzed by (**B**) SDS-PAGE and (**C**) Western blotting analysis. Lane M, protein molecular weight marker. (**D**) The purity of PfHDP in the fractions eluted by Ni-NTA and anion exchange chromatography was estimated by SDS-PAGE. Lane M, protein molecular weight marker; Lane 1, soluble fractions of the lysate; Lane 2, samples eluted from HisTrap^TM^ Ni-NTA column; Lane 3, samples eluted from HiTrap^TM^ Q HP anion-exchange column.

**Figure 2 nanomaterials-14-01440-f002:**
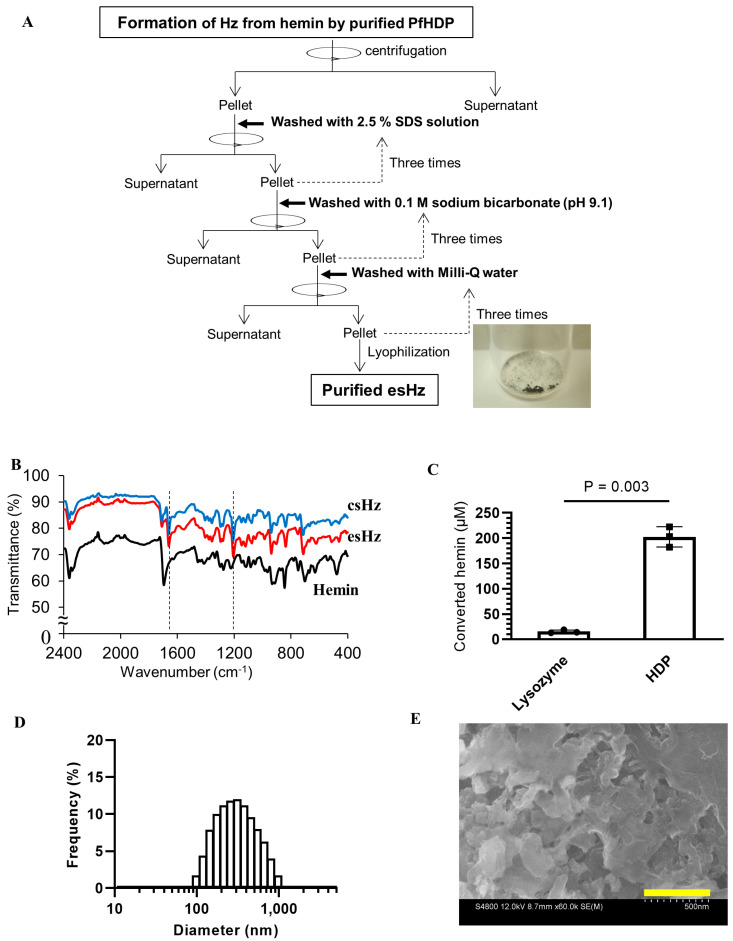
Qualification of an enzymatically synthesized analog of hemozoin (esHz) produced by recombinant PfHDP. (**A**) An overview of the formation of esHz from hemin by PfHDP and purification step. (**B**) The quality of esHz was estimated by measuring IR spectra. (**C**) The amount of converted hemin by PfHDP was calculated by monitoring the absorbance at 385 nm [4]. The Hz formation assay was repeated three times. Data are expressed as the mean ± standard deviation (*n* = 3). (**D**) Particle size distribution of esHz measured by dynamic light scattering analysis. (**E**) Scanning electron microscopy (SEM) image of esHz. Scale bars, 500 nm.

**Figure 3 nanomaterials-14-01440-f003:**
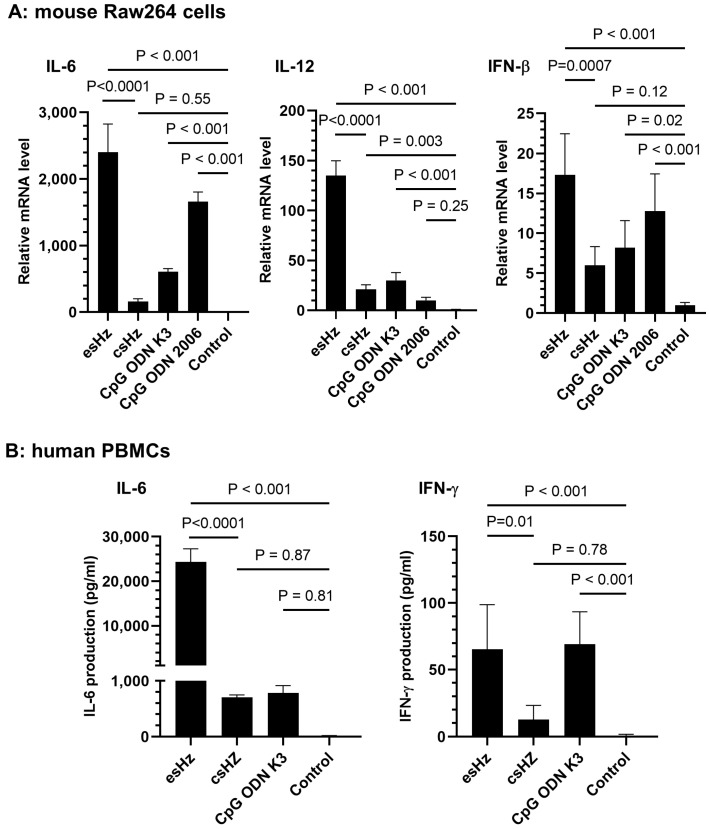
Cytokine induction by an enzymatically synthesized analog of hemozoin (esHz) from RAW264 murine macrophage cells and human peripheral blood mononuclear cells (PBMCs). (**A**) The induction of cytokines from RAW264 was evaluated using real-time reverse transcriptase-quantitative PCR (RT-qPCR) analysis. The Raw264 cells were incubated with esHz (50 µg/mL), chemically synthesized analog of Hz (csHz) (50 µg/mL), and with phosphorothioate-modified CpG ODNs, K3 and 2006 (1 µM) for 24 h. Data are expressed as the mean ± standard deviation (SD) (*n* = 5). The relative mRNA level compared to the control was calculated. (**B**) Immunostimulatory cytokine productions from human PBMC were evaluated using an enzyme-linked immunosorbent assay. Human PBMCs were stimulated with esHz (50 µg/mL), csHz (50 µg/mL), and phosphorothioate-modified CpG ODN (1 µM) for 48 h. Data are expressed as the mean ± SD (*n* = 5).

**Figure 4 nanomaterials-14-01440-f004:**
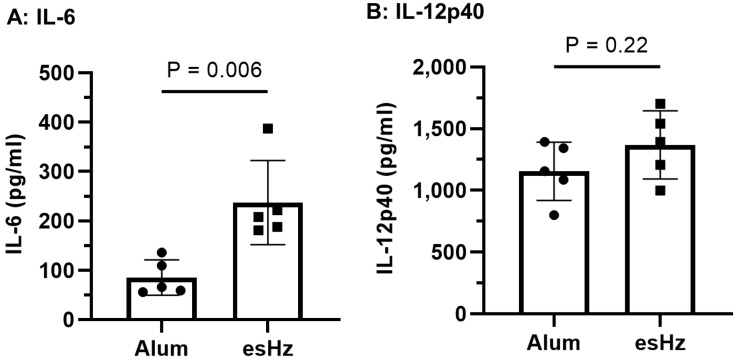
In vivo cytokine inductions in mouse serum by enzymatically synthesized analog of hemozoin (esHz). Interleukin (IL)-6 (**A**) and IL-12p40 (**B**) concentrations in mouse serum after intraperitoneal injection of esHz or alum. Mice received an injection of materials at a dose of 500 μg/mouse, and blood was collected at 4 h after the injection. Serum concentrations of IL-6 and IL-12p40 were measured using enzyme-linked immunosorbent assays. Results are expressed as the mean ± standard deviation of six mice. Data are representative of two independent experiments.

**Figure 5 nanomaterials-14-01440-f005:**
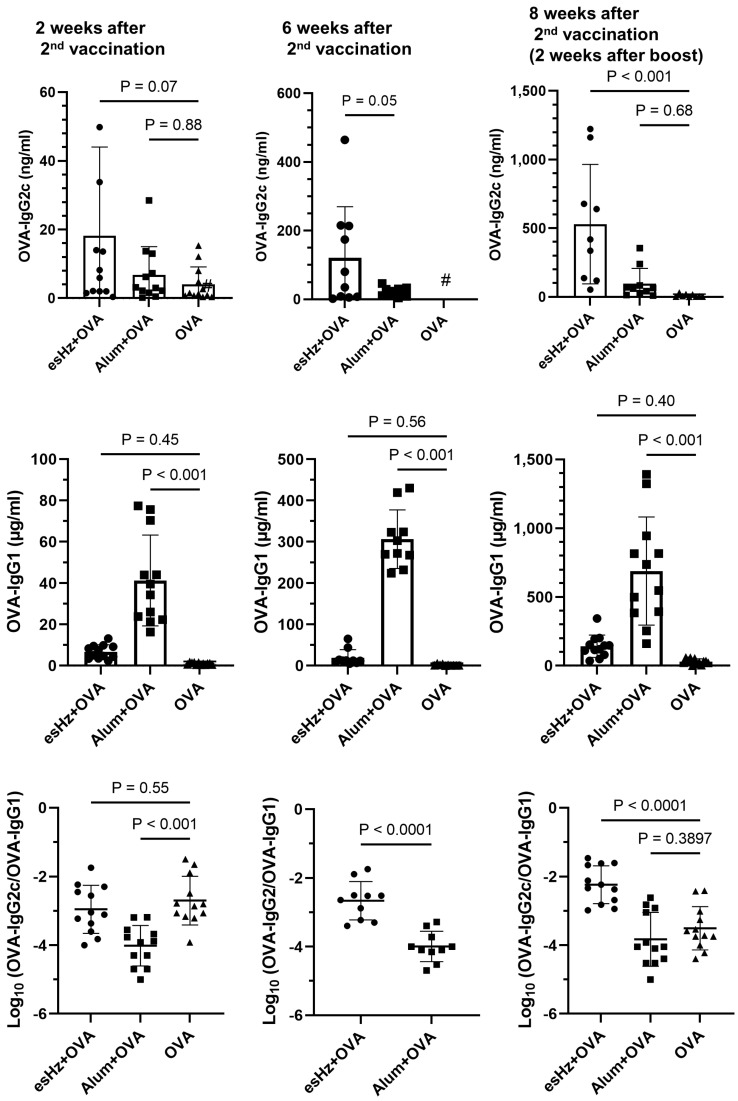
Ovalbumin (OVA)-specific IgG2c is induced by an enzymatically synthesized analog of hemozoin (esHz). The levels of OVA-specific IgG2c (OVA-IgG2c) and OVA-specific IgG1 (OVA-IgG1) and the ratio of OVA-IgG2c to OVA-IgG1 induced by either esHz or alum adjuvant at two (i), six (ii), and eight weeks (iii) after the second vaccination. Serum collected from immunized mice was used for enzyme-linked immunosorbent assay. Results are expressed as the mean ± standard deviation (*n* = 10 to 12). # The levels of OVA-IgG2c were lower than the detectable minimum of 0.39 ng/mL.

**Table 1 nanomaterials-14-01440-t001:** Primer sequence for analysis of murine cytokines.

Gene	Sequence (5′-3′)
GAPDH	FW: GTGGACCTCATGGCCTACAT RV: TGTGAGGGAGATGCTCAGTG
IL-6	FW: TCCTTCCTACCCCAATTTCC RV: CGCACTAGGTTTGCCGAGTA
IL-12	FW: GAAAGGCTGGGTATCGG RV: GGCTGTCCTCAAACTCAC
IFN-β	FW: GGTCCGAGCAGAGATCTTCA RV: TCACTACCAGTCCCAGAGTCC

FW: forward primer; RV: reverse primer; GAPDH: glyceraldehyde 3-phosphate dehydrogenase; IL: interleukin; IFN: interferon.

## Data Availability

The data that support the findings of this study are available from the corresponding author, T.Y., upon reasonable request.

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
