# Peer review of "Potential of Enzymatically Synthesized Hemozoin Analog as Th1 Cell Adjuvant"

_nanomaterials, 2024, doi:10.3390/nano14171440_

Round 1

Reviewer 1 Report

Comments and Suggestions for Authors

Hoshi and coworkers improved a method to generate heme detoxification protein (HDP), which in turn was used to enzymatically convert hemin to hemozoin (ezHz) at a reported efficacy of 34%. The immunostimulatory activity of ezHz was significantly higher than that of chemically synthesized (csHz) in terms of cytokine production by a murine macrophage-like cell line and human PBMC. When employed as an adjuvant for immunization, in contrast to Alum ezHz induced a Th1-biased antibody response in mice.

The study needs to address several issues.

Major points:

1. Since HDP is generated in E. coli it is possible that the protein contains endotoxin, which has frequently been reported to exert strong immunomodulatory effects. The authors need to detect endotoxin in the ezHz preparations used for subsequent cellular and in vivo studies, since it remains possible that part of the biological activity of esHz was due to an according contamination. 

2. Numerous reports have demonstrated that hemin exerted strong activation of immune cells (PMID: 15917143). In this study, HDP converted only 34% of hemin into hemozoin. It is not clear from the manuscriupt whether hemin was separated from the reaction. The authors need to comment on whether the mixture of hemin and hemozoin has been used for the functional assays, and whether this may explain in part the differences in functional activities between ezHs and csHs.

Minor points:

1. Figure 3: The authors should also show statistical significancies of either group versus Control.

2. Figure 4: The legend as well as the accoding part of the materials and methods section (2.8) state that cytokines were detected 2 hours after injection. This time point is very early and virtually excludes stimulation-induced transcriptional alterations in gene expression. Cytykine analysis after 6h and 24h, respectively, are more common. The authors nedd to comment on that.

3. Figure 5: Please indicate statistical significant differences between either group and `OVA´.

4. Line 383: Please correct "con-troversial3".

5. Please delete the last two sentences of the Discussion.

Reviewer 2 Report

Comments and Suggestions for Authors

The manuscript focuses on an enzymatic production method of heme using heme detoxifying protein (HDP), which catalyzes the formation of heme dimers in Plas mod. We investigated the immunostimulatory effects of hemozoin analogs (esHz) synthesized by enzymes produced by recombinant Plasmodium falciparum HDP (PfHDP). ESHz stimulates macrophage cell lines and human peripheral monocytes, leading to the production of interleukin-6 (IL-6) and IL-12p40. In mice, subcutaneous administration of esHz and antigen ovalbumin (OVA) increased the level of OVA specific immunoglobulin (Ig) G2c isotype in 33 serum samples, but did not induce OVA specific IgG1. This manuscript needs significant improvement in the following areas:

(1) The characterization of the product is too simple to confirm the correctness of the compound, and it is necessary to increase the sequencing data of mass spectrometry and its corresponding structure. FT-IR needs to be further improved, and the overall paper lacks some key data characterization

(2) The manuscript contains too many errors in writing and requires significant improvement

Comments on the Quality of English Language

Some expressions have errors, especially in terms of units and subscripts

Reviewer 3 Report

Comments and Suggestions for Authors

The manuscript "Potential of enzymatically synthesized hemozoin analog as Th1 cell adjuvant"  is well written.

1. The introduction contains abundant intricate technical information, which can make it challenging for readers not well-versed in the subject matter to comprehend the context promptly. Please consider reducing these statements or dividing them into smaller, more easily comprehensible segments.

2. There are numerous comma splices and run-on sentences. For instance, the language in lines 31-33 might be dissected to enhance its intelligibility.

3. There are typographical errors, such as "obtined" instead of "obtained" on line 89.

4. The sections that explain the HDP expression and purification (lines 95-123) contain a substantial amount of information. It is advisable to utilise bullet points or numbered lists to divide steps into smaller parts. For example, "HDP was first identified in Plasmodium falciparum and orthologs have now been found in seven other species of Plasmodium [2]" can be split into two sentences.

5. Ensure that abbreviations and terminology are used consistently throughout the manuscript. For instance, both "hemozoin" and "Hz" are employed; it is essential to provide a clear definition of abbreviations when they are first used and consistently utilise one term after that.

6. Ensure that all figures and tables are clearly labeled and referenced in the text. They should also be of high quality and easy to interpret.

7. The document contains placeholders for citations ("[1]", "[2]", etc.) that require updating with the appropriate references. Additionally, be sure that the citation format adheres to the specific guidelines set by the publication.

I recommend a minor revision.

Round 2

Reviewer 2 Report

Comments and Suggestions for Authors

At present, the manuscript has basically solved the problem raised for the first time, therefore it is agreed to be recommended for publication in this journal